# PROCEEDINGS A

meteorology

flood, economic cost, input–output analysis, indirect costs, hypothetical extraction model, York (UK)

**Author for correspondence:**
Dabo Guan
e-mail: dabo.guan@uea.ac.uk

# Assessing the economic impacts of IT service shutdown during the York flood of 2015 in the UK

Yang Xia[1], Dabo Guan[1,2,3], Albert E. Steenge[4], Erik Dietzenbacher[4], Jing Meng[5] and David Mendoza Tinoco[6]

[1]Water Security Research Centre, School of International Development, University of East Anglia, Norwich NR4 7TJ, UK
[2]Department of Earth System Sciences, Tsinghua University, Beijing 100080, People's Republic of China
[3]School of Management and Economics Beijing Institute of Technology, Beijing 100081, People's Republic of China
[4]Faculty of Economics and Business, University of Groningen, Nettelbosje 2, 9747 AE Groningen, The Netherlands
[5]The Bartlett School of Construction and Project Management, University College London, London WC1E 7HB, UK
[6]Economic Analysis Program of Mexico, Centre of Economic Studies, The Collage of Mexico, Mexico

JM, 0000-0001-8708-0485

In this paper we focus on the 'Christmas' flood in York (UK), 2015. The case is special in the sense that little infrastructure was lost or damaged, while a single industry (IT services) was completely knocked out for a limited time. Due to these characteristics, the standard modelling techniques are no longer appropriate. An alternative option is provided by the Hypothetical Extraction Method, or HEM. However, there are restrictions in using the HEM, one being that no realistic substitutes exist for inputs from industries that were affected. In this paper we discuss these restrictions and show that the HEM performs well in the York flood case. In the empirical part of this paper we show that a three-day shutdown of the IT services caused a £3.24 m to £4.23 m loss in York, which is equivalent to 10% of the three days' average GVA (Gross Value Added) of York city. The services sector

(excluding IT services) sustained the greatest loss at £0.80 m, where the business support industry which was predominantly hit. This study is the first to apply a HEM in this type of flood on a daily basis.

## 1. Introduction

Flooding was widespread in the UK during the 2015 Christmas season, putting a large number of roads, railways, houses and buildings at risk. The most severe flooding occurred on the night of Christmas Day (25 December 2015) and lasted until 27 December 2015 (the day after Boxing Day [1]). The city of York was hard hit by the flood, where houses and businesses in the city centre in particular experienced severe flooding after the banks of the River Ouse burst [1]. The flooding led to a broad IT service shutdown and knocked out the power to the British Telephone (BT) York exchange. Broadband cables were also damaged by flood water in the York BT exchange. As a consequence, thousands of York homes and businesses experienced phone and broadband services outage (The Guardian, 28 December 2015). Shops could not accept card payments and cash machine services from Natwest, Lloyds and Yorkshire Bank were out of order (The Guardian, 28 December 2015). As broadband services are usually physical products that are mostly provided by local service carriers, they cannot readily be substituted by services from elsewhere. As a result, the IT outages disrupted almost all commercial transactions and economic activities in shops, restaurants and supermarkets of York which relied heavily on digital payment methods for three days during the Christmas and Boxing Day period. A news journalist described York during those days as a 'ghost town' (The Guardian, 28 December 2015).

Traditional ways of flood and disaster modelling become less useful here for several reasons. The most important is that existing flood and disaster modelling, such as approaches based on recently presented adaptive models [2–4], heavily rely on quantifying the damage to infrastructures as a direct and tangible consequence of flooding. However, there was not that much damage to the infrastructure in the York flood, which makes it difficult to implement standard ways of disaster modelling. Instead, the flood induced substantial indirect and intangible costs from the IT service shutdown, where no direct alternatives were available. As York is a core commercial hub of the region of Yorkshire and the Humber, an IT service blackout can seriously affect upstream and downstream industries that rely on those services, particularly so during the busy Christmas season where the timing of the flooding undoubtedly exacerbated its economic impact. The flood knocked out the IT services in York for exactly three days, without any adaptive processes being available as substitutes. Both points imply the need for a more appropriate approach to better fit the distinctive characteristics of the York flood.

In this paper we apply the Hypothetical Extraction Method (HEM) by perceiving the three days' IT shutdown as the 'extraction' of the IT services industry from the York economy for that period. A HEM is able to measure the overall reduction in the production level after extracting selected industries from the economy. For flood and disaster research, the method becomes an option when some industries partially, or completely, are disconnected from other industries. Below, we discuss the HEM in the context of the York flood in 2015 to determine the total indirect economic loss. The HEM was structured on a daily basis because the flood induced exactly three days of shutdown of the IT services industry in York. We show that our approach is able to quantify the economic impact on the city-level gross value added (GVA) when an industry/industries is/are hypothetically extracted from the economy in question. The next section starts with a brief review of the existing literature on assessing flood risks. Section 3 describes the key methodologies adopted in this study and their relevance. Section 4 discusses the main results of the study and the paper ends with a conclusion and suggestions for policy implications in §5.

## (a) The relevance of small and special cases

Before continuing, it may be useful to address one particular question, i.e. is it worthwhile to pay attention to what some may call a very small and very special case. Addressing this question, clearly, depends on what we mean by a 'special case'. The case we are dealing with concerns one industry, a flood of (only) three days and the Christmas period. In this sense it is indeed a very special case, but it did happen in the real world. In fact, in our view, this flood offers a unique challenge to theorists and data gatherers to test the available models and to come up with the most appropriate framework to analyse this case. It is true that it is a small example in terms of size and consequences, but our message is that this is a case where hypothetical extraction does fit very well.

Regarding 'size', it is true, of course, that, in general, the literature tends to pay attention to large disruptions. However, a small disruption in a single firm in one place may cause an enormous effect somewhere abroad or disrupt an entire industrial network [5]. A now famous example described in Latour [6] is the small 10-minute fire at a Philips semiconductor plant in March 2000 in Albuquerque, NM. The production of computer chips came to a halt and it took Philips a couple of weeks to restore the plant's capacity. However, that resulted in a potential 400 million USD revenue impact at Swedish company Ericsson. Another, more recent discussion from Sheffi [7] is that due to the interconnection along the production supply chain in car manufacture, any risk—such as corporate social responsibility, cybersecurity, or global raw material risks that affect a single supplying sector—will eventually affect the car companies due to their sector interdependencies (see [7]).

## 2. Literature review

## (a) The input–output model in modelling the impact of floods

A flooding event has both direct and indirect effects. Direct effects include injury and damage to physical assets [8], while indirect effects are secondary effects resulting from a flooding event including reconstruction afterwards [3,4,9–13]. This paper follows the definitions in disaster parlance, where the 'direct economic impacts' are defined as the initial loss in commercial and private buildings, machinery and materials as well as the resulting reduction in output level within the IT service industry, while 'indirect economic impacts' denote the secondary and cascading loss in all other industries from industrial interdependencies in York city [14].

Variants of the basic input–output (IO) model have been widely applied in flood risk analysis. Schulte in den Bäumen et al. [15] applied IO analysis to quantify the cascading effects on the production possibilities of Germany's federal states and foreign economies resulting from the 2013 flooding disaster. In their work, the role of reduced production possibilities and the interconnections along production supply chains have been highlighted in post-disaster risk estimates. Among other flood risk analysis, studies of the recovery to the pre-disaster situation have been prominent. Steenge & Bočkarjova [16] proposed an 'imbalanced' recovery model based on an IO analysis to assess the economic impacts of a flood. Hallegatte [2] and Hallegatte et al. [3] integrated the dynamic nature of post-disaster economic recovery into adaptive regional IO (ARIO) models by relating overall production capacity to the adaptive behaviour of businesses and households during the disaster's aftermath. Koks et al. [11] employed the imbalanced recovery model to reflect production loss in the post-disaster economy and then applied the ARIO model to simulate the economic recovery period for the Rotterdam (the Netherlands) harbour area. Very recently, Mendoza-Tinoco et al. [12] proposed the term 'flood footprint' to denote the total socioeconomic impact of a flooding event on both the flooded area and the entire economic system, as measured by the total cumulative economic loss until the economy is fully recovered.

One particular aspect of modelling post-disaster situations is that there are special cases. 'Normal' floods will either affect physical infrastructure or impact industries differently in terms of their production capacity. In contrast, the York flood completely 'paralysed' the IT services

industry for three days without much direct impact on other industries. Therefore, the nature of the York flood differs considerably from a 'normal' flood. This also means that accepted ways of flood modelling, such as employing adaptive modelling or the Inoperability Input–Output model (IIM), which generally emphasize or heavily rely on quantifying the production capacity loss in industries, cannot be used effectively. In fact, contrarily to many 'standard cases', the York flood caused little damage to physical capital but severe damage to 'soft' services, in particular the IT services, which implies the need for a different way of modelling economic loss. In this respect, the HEM, which we shall briefly introduce in the next section, is potentially a suitable candidate.

There is another aspect that should be noticed here. In recent publications, a number of exogenous parameters have been introduced that determine the total flood impact. One such exogenous variable is 'recovery time'. Koks & Thissen [17] showed that the value given to this variable may be the most important determinant of the entire study, which may be problematic because then there is no explanation for a variable determining the scale of the disaster; see also Mendoza-Tinoco *et al*. [12] on this.

However, in the York case the length of recovery period was exactly determined. In this respect, our current study provides an interesting case, in which such a core parameter, the recovery period, is absent due to the flood characteristics.

## (b) Applications of the hypothetical extraction method

The HEM method was first proposed to estimate the relative importance of certain industries for an entire economy. This was done by introducing the concept of hypothetical extraction of the industry, thereby assuming that the interruption of its domestic services was remedied by imports. The reduction in overall domestic production level after extracting a certain industry defines the importance of the industry. The method was first introduced by Paelinck *et al*. [18] and Strassert [19], and later re-formulated by Meller & Mafan [20] and Cella [21]. Once an industry is hypothetically eliminated from the economic system, the HEM can be used to estimate the effects of this extraction on other industries and on the wider economic system. Thus, the difference between the output level of the other industries before and after the extraction reflects the linkages between the extracted industry and the rest of the economy, where these linkages can be further decomposed into backward and forward linkages [22]. Backward linkages refer to the linkages between an industry and other industries that supply inputs to it while forward linkages refer to the linkages between an industry and other industries that purchase output from it [23].

Recently, the HEM has been reformulated again by several researchers, including Miller & Lahr [24], Dietzenbacher & Lahr [25] and Temurshoev & Oosterhaven [26]. For example, Miller & Lahr [24] reformulated the HEM by extending it from the quantity side to the cost side of the economy. By doing so, they are able to identify the cost burden interdependencies among economic sectors. With the advantage of capturing the relative magnitude of an industry's final demand and its relative effects on the gross production level when compared to the traditional multiplier method, the original application of the HEM has been broadened to include environmental impact analyses. Ali [22], for example, analysed the direction and strength of the relationship between linkages among industries and their contributions to $CO_2$ emissions in Italy. Similarly, Zhao *et al*. [27] investigated sectoral $CO_2$ emission linkages in China at the regional level by integrating the HEM with a multi-regional IO (MRIO) model. With increasing focus on global climate change, the HEM has also been applied in resource studies, including water resources [28–30] and energy use [31].

Two further points are particularly worth consideration when applying the HEM. Firstly, there are several variants. Therefore, it is important to determine the specific variant of the HEM to be used in the study and before applying it. Following Miller & Blair [23], a column of an industry in an IO table should be replaced by a column of zeros if it cannot buy any intermediate inputs from other industries; the backward linkages of this industry no longer exist. Analogously, a row of an industry in an IO table should be replaced by a row of zeros if it has no intermediate sales to other industries and its forward linkages no longer exist. Both should be replaced by zeros if both

backward and forward linkages of an industry cease. Secondly, it is also crucial to decide how large a percentage of an industry's backward and forward linkages should be reduced. In other words, whether the industry should be eliminated completely or partially [25] if the total capacity of the industry is put out of work. In this respect, there are two main approaches to implementing the HEM. Following the original HEM, as developed by Strassert [19] and implemented by Schultz [32], 'extraction' simply means completely removing the backward and forward linkages of an industry or replacing its row and column elements with zeros in the input coefficient matrix. Alternatively, Cella [21] improved the original extraction method by differentiating economic activities across all industries into two categories: intermediate sales and purchases with other industries and self-reproducible sales and purchases. Thus, an extracted industry no longer sells or purchases any intermediate products to or from other industries, and its technical coefficients will be partially replaced with zeros while the others remain the same [22,27]. Although such an economic assumption seems intuitively unrealistic because technical coefficients are linked to each other, Dietzenbacher & van der Linden [33] validated this assumption by introducing imports to sustain the original technical production process.

In a recent paper, Oosterhaven [34] put forward an important 'caveat' regarding the use of the HEM in disaster studies. The core of his criticism concerned the use of zeros in horizontal rows of the matrix of input coefficients (see §3). These zeros can be problematic if substitution possibilities exist between domestic and foreign deliveries. In our view, this criticism strongly supports the use of the HEM in cases like ours where IT services were cut off almost completely without substitution possibilities. In standard HEM applications that consider a longer term, the (often implicit) assumption is that the extracted inputs are substituted by imported inputs. The technical production process does not change in the sense that all inputs that were required previously are still necessary. In the current ultra-short-term period of three days, it seems plausible to assume that the other industries manage to produce without purchasing IT services. If these domestic IT services are not substituted by imported IT services, this implies that the value-added coefficients will increase, for example, to pay for overwork.

Our case also provides an answer to Oosterhaven's view of the method [34]. In his criticism, Oosterhaven points out that there is a difference between applying HEMs to backward and to forward effects. Using HEM for studying the impacts of upstream, backward effects is correct and poses no problem in terms of interpretability. However, interpreting the extraction of a row of the coefficients matrix to represent the forward, downstream impacts of the extracted industry is faulty [34, p. 8] because 'it only measures the direct impacts of the complete disappearance of the demand for an industry's intermediate sales' [34, p. 8]. What Oosterhaven [34] basically argues is that the extraction behaviour in the HEM method is based on the assumption that all downstream demand for a sector's intermediate sales simply and completely disappeared after the incident. In contrast, he argues that such downstream demand for intermediate sales will not disappear, but rather, seeks for substitutions elsewhere. Also, the HEM neither measures the higher order backward impacts of this disappearance, nor the forward impacts of the secession of these sales upon the purchasing industries. This is, of course, the case for any disaster that lasts longer and when sectors have enough time to adapt. However, we suggest his criticism does not hold for our special York flood case. This is because the IT outage resulting from the flood only lasts for three days, which provides insufficient time for economic sectors to adapt or seek other substitutions. This is also because of the special nature of IT services produced by local carriers that invalidate the replacement or substitutions from imports. Therefore, in our view, Oosterhaven is right in his judgement in standard cases. However, in the case of a complete shut-down of a particular industry, including its transport and transmission functions, the use of zeros—as we have put forward—is allowed. We witness the 'complete disappearance' of the supply and demand mechanism of the products of the industry in question while no substitution or replacement takes place (because of the special nature of the industry). However, it is worth noting that a HEM can only be used with a focus on backward linkages. Setting coefficients in the column at zero is equivalent to removing the backward dependence of a 'removed' industry on other industries, while setting the row coefficients to zero means eliminating the backward

dependence of the other industries on this particular industry. Neither action can deal with the downstream effects. Therefore, the calculations of the following empirical case only include backward effects, indicating that if IT services are blocked, they would no longer require inputs and these inputs would no longer need to be produced.

In this context, a further question needs considering. This question concerns the possibility that part of the costs will be recovered in the days after the three-day shut down. This is true, clearly, for the costs associated with any type of disaster, whether the shutdown takes days, weeks or even longer. The point here is that the shutdown causes various types of damages, each of which must be dealt with after the disaster. So the local/regional economy must, somehow, organize itself to address these in the weeks after, in this case, the flood. Focusing too much on this point carries the danger of concentrating too much on the size of the disaster, which—in our view—is not appropriate in this case. Moreover, the same argument can be used for 'large disasters', after six months (or 18 months, or five years) we are back to usual. For a discussion, in this context, of the role of resilience increasing re-scheduling methods, see the recent contribution of Park *et al.* [35]. A related type of argument, which looks in particular at post-disaster technological changes and adaptations can be found in Park *et al.* [36] and Xei *et al.* [37]. For identifying the regional economic impacts of a big disaster (9/11) in an overall context of resilience-related change, see Park *et al.* [38].

The proposed method tries to quantify, as effectively as possible, the associated costs in certain numerical figures. The interesting point here is that the flood—not a laboratory case, but something that happened in the real world—precisely fits the conditions for applying the HEM method. The recent literature is quite clear on these conditions, and the York flood offers a most interesting real case story here.

# 3. Methodology

## (a) The input–output model

The traditional IO model is based on the assumption of a one-to-one relationship between an industry and its characterizing product. That is, each industry produces a distinct commodity that can be used for either final demand by sub-categories such as households, governmental agencies, exports or capital formation, or for intermediate demand by other industries. The total output $x_i$ of industry $i$ in an $n$-industry economy is:

$$x_i = z_{i1} + \cdots + z_{ij} + \cdots + z_{in} + f_i = \sum_{j=1}^{n} z_{ij} + f_i, \tag{3.1}$$

where $z_{ij}$ stands for the value of transactions from industry $i$ to $j$, $f_i$ for the final consumption for industry $i$. In matrix notation, we have,

$$\mathbf{x} = \mathbf{Z}\mathbf{i} + \mathbf{f}, \tag{3.2}$$

where $\mathbf{x}$ is the $n \times 1$ vector of total industrial outputs, $\mathbf{f}$ the $n \times 1$ vector of final demands, $\mathbf{Z}$ the $n \times n$ matrix of interindustrial deliveries and $\mathbf{i}$ a vector of ones. The model is standard given in technical coefficients $a_{ij}$ which are defined as

$$a_{ij} = \frac{z_{ij}}{x_j}. \tag{3.3}$$

Standard $a_{ij}$ is taken as fixed in the short run. Combining equations (3.1) and (3.3), the basic Leontief IO model can be written in a matrix term as equation (3.4).

$$\mathbf{x} = \mathbf{A}\mathbf{x} + \mathbf{f}. \tag{3.4}$$

Solving for $\mathbf{x}$, we have

$$\mathbf{x} = (\mathbf{I} - \mathbf{A})^{-1}\mathbf{f} = \mathbf{M}\mathbf{f}, \tag{3.5}$$

where $\mathbf{M} = (\mathbf{I} - \mathbf{A})^{-1}$ is the Leontief inverse or multiplier matrix. The elements $m_{ij}$ stand for both the direct and indirect output requirements of an industry $i$ to produce one unit of final consumption in an industry $j$.

## (b) The hypothetical extraction method

Written out, the Leontief IO model holds as in equation (3.6), where the technology as represented by matrix $\mathbf{A}$ is given, final demand $\mathbf{f}$ is determined exogenously, and output $\mathbf{x}$ endogenously. We have, written out,

$$\begin{pmatrix} x_1 \\ \vdots \\ x_n \end{pmatrix} = \begin{pmatrix} a_{11} & \cdots & a_{1n} \\ \vdots & \ddots & \vdots \\ a_{n1} & \cdots & a_{nn} \end{pmatrix} \begin{pmatrix} x_1 \\ \vdots \\ x_1 \end{pmatrix} + \begin{pmatrix} f_1 \\ \vdots \\ f_n \end{pmatrix}. \tag{3.6}$$

Suppose now that industry 1 ceases production due to a major catastrophe. Consequences can be modelled if we follow the HEM concept by completely extracting industry 1 from the economy. This means that there will no longer be any intermediate transactions with the other industries. This extraction can be achieved by simply removing its backward and forward linkages with other industries and itself. Thus, the extracted $n \times n$ matrix turns into a new technical coefficient matrix $\mathbf{A}'$ with first row and first column equal to 'zero', i.e.

$$\mathbf{A}' = \begin{pmatrix} 0 & \cdots & \cdots & 0 \\ \cdots & a_{22} & \cdots & a_{2n} \\ \vdots & \vdots & \ddots & \vdots \\ 0 & a_{n2} & \cdots & a_{nn} \end{pmatrix}. \tag{3.7}$$

A new final demand vector may arise when a major catastrophe alters the patterns of household and government consumption. Households may spend more on life necessities and less on luxury and entertainment while government may spend more on reconstruction and health care. However, the current study considers neither changes in final demand or imports. There are several reasons for this. Firstly, the time period in consideration is relatively short and is most likely insufficient for consumers and government to react in a way that changes their consumption behaviour. Secondly, although outages of some products can sometimes be compensated by imports, IT services are generally provided by local carriers in York. Therefore, there is no immediate import available for IT services during the three-day outage and, thus, the IT services cease to exist temporarily, which implies that there are no outputs nor deliveries to final users (see also §3c). We thus have

$$\begin{pmatrix} x'_1 \\ \vdots \\ \vdots \\ x'_n \end{pmatrix} = \begin{pmatrix} 0 & \cdots & \cdots & 0 \\ \cdots & a_{22} & \cdots & a_{2n} \\ \vdots & \vdots & \ddots & \vdots \\ 0 & a_{n2} & \cdots & a_{nn} \end{pmatrix} \begin{pmatrix} x'_1 \\ \vdots \\ \vdots \\ x'_n \end{pmatrix} + \begin{pmatrix} f'_1 \\ \vdots \\ \vdots \\ f'_n \end{pmatrix}, \tag{3.8}$$

where $\mathbf{x}'$ is the new output level while $\mathbf{f}'$ is the new final demand for the correspondingly reduced final-demand vector. If $f'_1 = 0$ and $f'_j = f_j$ for all other $j$, $x'_1 = 0$. (see further [23], Ch 8). In matrix notation, the difference in total output then is obtained as

$$\mathbf{x} - \mathbf{x}' = (\mathbf{I} - \mathbf{A})^{-1}\mathbf{f} - (\mathbf{I} - \mathbf{A}')^{-1}\mathbf{f}'. \tag{3.9}$$

## (c) Deriving a city level input–output table from a national table

Firstly, due to the lack of city-level technical coefficients, we derived the regional technical coefficients for Yorkshire and the Humber from the UK national IO table using the Flegg and Webber scaling-down approach. Then, the obtained regional technical coefficients were applied to York in the current study by assuming that the city has the same technical coefficients as Yorkshire

and the Humber. IO tables are traditionally developed at the national level by the relevant statistical bureaus. This also is the case for the UK where Supply and Use tables, the building blocks of IO tables, are produced yearly by the Office for National Statistics. This paper used the Augmented Flegg Location Quotients (AFLQ) technique to obtain the regional IO coefficients matrix for Yorkshire and the Humber from the national IO statistics. Regional economies, clearly, can substantially differ from national economies in terms of trading relationships. Furthermore, intermediate purchase from other regions should be regarded as a leakage under a regional economy but as domestic production under a national economy. For this purpose, economic data on employment of the regional economy is used to re-scale the national coefficients to better reflect the regional economic structure of Yorkshire and the Humber. The process consists of adjusting the national coefficients to the regional scale by measuring the relative size of each industry for the regional economy in relation to the relative size of the same industry for the national economy; adjusting by certain parameters to consider the commercial traffic between the regional economy and other regions, and the possible specialization of an industry within the region.

The regional input coefficient, $r_{ij}$, is then derived from the national technical coefficients, $a_{ij}$, when re-sized by a regional-economy parameter or *location quotient*, $lq_{ij}$, as in equation (3.10):

$$r_{ij} = lq_{ij} * a_{ij}, \tag{3.10}$$

where $r_{ij}$ is the amount of input from industry $i$ needed to produce one unit of output in industry $j$. We start from the so-called simple location quotients (SLQ), to assess the relative importance of each regional industry $i$, as described in equation (3.11).

$$SLQ_i = \frac{RE_i/TRE}{NE_i/TNE} \equiv \frac{RE_i}{NE_i} * \frac{TNE}{TRE}, \tag{3.11}$$

where TRE is total employment in the region, TNE total employment in the country, $RE_i$ regional employment in industry $i$, and $NE_i$ national employment in the same industry.

The cross-industry LQ (CILQ) has been derived from the SLQ to assess the relative importance of a supplier industry $i$, regarding the purchasing industry $j$ as in equation (3.12):

$$CILQ_{ij} = \frac{RE_i/NE_i}{RE_j/NE_j} \equiv \frac{SLQ_i}{SLQ_j}. \tag{3.12}$$

As intermediate sales between regions were often treated as domestic production in the CILQ, it will underestimate the regional imports. In a later contribution, Flegg & Webber [39] refined the regionalization in the Flegg LQ (FLQ) to correct for the persistence of underestimation of regional imports in the CILQ through the parameter λ to obtain the FLQ in equations (3.13) and (3.14):

$$FLQ_{ij} = CILQ_{ij} * \lambda, \tag{3.13}$$

where

$$\lambda = \left[1 + \frac{TRE}{TNE}\right]^{\delta}, \quad 0 \leq \delta < 1. \tag{3.14}$$

The final version of the location quotient used in this paper is the Augmented FLQ, which includes the term $[\log_2(1 + SLQ_j)]$ for including the effects of regional specialization:

$$AFLQ_{ij} = CILQ_{ij} * \lambda * [\log_2(1 + SLQ_j)]. \tag{3.15}$$

Note that the effect of applying the logarithmic transformation to $SLQ_j$ is that a larger region now is more likely to be allocated a bigger allowance for regional imports than a smaller region [40].

Secondly, the data for final consumption for the 45 industries in Yorkshire and the Humber were scaled down to city-level data for York based on the city-to-regional GVA ratio in 2015, which was calculated as 4.7%. Based on the final consumption data for York, we derived the city-level IO table (for York) by assuming it has the same technical coefficients as York and the Humber region (equations (3.4) and (3.5)). Then, the data for the aggregated final consumption

vector is obtained. Subsequently, the 45 York industries were divided by 365 to obtain the daily value of each industrial final consumption, and the results were multiplied by three to calculate the industries' total final consumption during the three-day shutdown period of the IT services industry. We assumed that final consumption did not change during the flooding because of the relatively short time period for households and the government to react.

## (d) A hypothetical extraction model for York in 2015

In view of the above, the IT service industry can be treated as a 'key' sector in the economy of York when considering its domestically supplied inputs [23]. The extraction of the IT service industry's linkages with other industries follows the original HEM. As a result, in the current study, both the backward and forward linkages of the IT services industry were eliminated (set to '0') in the technical coefficient matrix, representing a complete blackout of IT services. Then, the newly obtained technical coefficient matrix—with a '0' column and row for the IT services industry and data for three days' worth of industrial final consumption in York—was used in equation (3.6) to calculate the new industrial output level required to support three days' final consumption. Finally, this new output level, without IT service support, was compared with the original output level for satisfying three days of final consumption with IT services in place.

In order to consider the excessive transaction volumes during the Christmas shopping period, we provided an upper bound for the results following the same methods but employing different values for final demand during the three-day IT outages. Due to the lack of daily sales data for York, we assumed the same monthly trend in household expenditure as the UK. According to data from the Office for National Statistics [41], household expenditure on food, drink and tobacco, clothing and footwear and other household goods during December are 16, 42 and 31% higher than those of other non-Christmas monthly sales during 2015. Therefore, we adjusted in this way the original three days' final demand that was calculated from the IO table by using the excessive sales value to reflect the excessive final demand during the Christmas period in York.

## 4. Results

After a three-day shutdown of IT services, the proposed HEM revealed total economic losses of £3.24 million, of which the IT services industry suffered the largest part, £1.83 million (56.48%); the remaining £1.41 million (43.52%) in losses were distributed among the remaining 44 industries. This result is expected to rise to £4.23 million when considering the excessive volumes of transactions during the Christmas season. The results re-emphasize the importance of considering indirect economic losses in disaster risk assessment, as over 40% of the economic losses in the present case resulted from the cascading effects of industrial inter-dependencies. As pointed out above, excluding the IT services industry, the remaining industries suffered a total of £1.41 million in economic losses due to the three-day shutdown of the IT industry. This substantial loss resulted from the interdependencies between these industries and the IT services industry. Unsurprisingly, the services sector suffered the greatest economic losses among the three broad sectors (£0.80 million), accounting for 57% of the total economic losses of £1.41 million (not including the economic losses suffered by the IT services industry), see figure 1. The chart shows the proportions of economic losses in three broad sectors, namely, manufacturing, services sectors and other (agricultural and mining, energy supply and construction). Percentages are displayed inside the circle.

According to the 2011 Census in the city of York from Neighbourhood Statistics [42], the local economy is mainly led by the service sector including Wholesale and Retail Trade, Human Health and Social Work Activities, Education and Accommodation and Food Service Activities. Agriculture, Forestry and Fishing, Mining and Quarrying (all included in Other) as well as Manufacturing occupy only a small portion of the economy. Therefore, we specifically focus on the economic loss occurred in the service sector in York. Among the 25 industries in that sector, the business support services industry sustained the greatest indirect economic losses

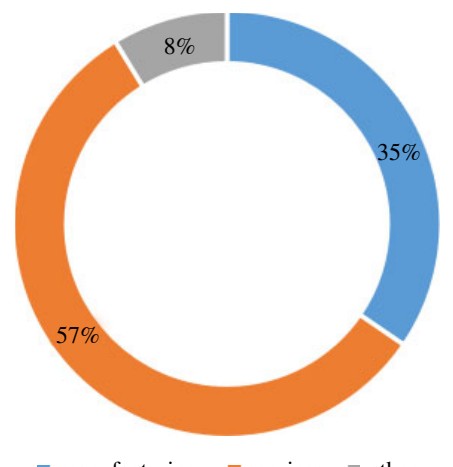

economic losses in three broad sectors (million £)

■ manufacturing  ■ services  ■ other

**Figure 1.** Economic losses in three broad sectors. (Online version in colour.)

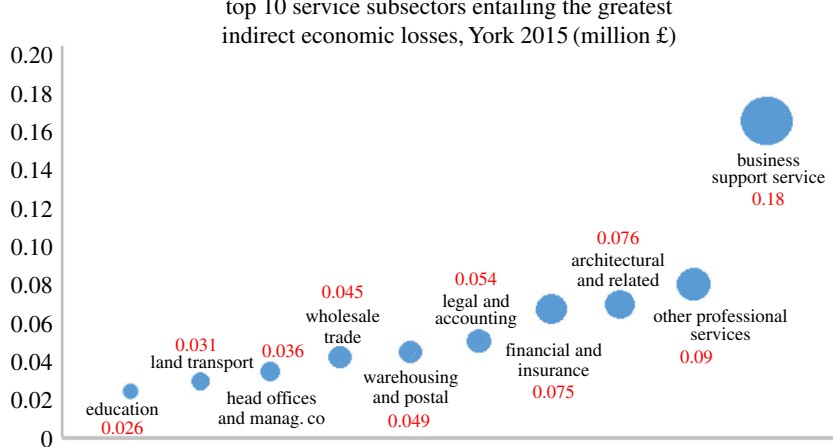

top 10 service subsectors entailing the greatest
indirect economic losses, York 2015 (million £)

**Figure 2.** The top 10 service industries suffering the greatest indirect economic losses in York, UK in 2015. (Online version in colour.)

from the IT service shutdown (£180 k), followed by the other professional services industry (£90 k) (figure 2). Additionally, the financial and insurance (£75 k), architectural (£76 k), legal and accounting (£54 k), warehousing and postal (£49 k), wholesale trade (£45 k) and head offices and management (£36 k) industries were also negatively affected by the IT service shutdown, see figure 2.

The diagram shows the 10 service industries suffering the greatest economic losses among the 45 total industries in York in 2015 due to the three-day IT service shutdown resulting from the 2015 York flood. The different industrial indirect economic losses originate from their different levels of dependency on the IT services industry. The y-axis shows the value of economic losses measured in millions of dollars and the spot sizes represent the different magnitudes of economic loss in each industry.

Our findings show that a number of manufacturing industries were also affected by the shutdown of the IT services, with an overall loss of £0.49 million, accounting for 35% of the total non-IT services sector economic losses (figure 1). Among the 15 manufacturing industries, the computer industry, likely due to its close relationship to IT services, suffered

the greatest losses (£90 k), followed by other manufacturing (£63 k), metals (£55 k) and non-metallic (£53 k) industries. Other industries, including agricultural and mining, energy supply and construction industries stand for 8% of the non-IT economic losses. These results reveal the heavy reliance of the supply of IT services in their operations and production. This, on the other hand, also acknowledges and confirms the recent remarkable growth in IT outsourcing in the UK. Miozzo & Grimshaw [43] reported that large service firms, client organizations and manufacturing companies in the UK have substantially outsourced IT services to multinational technology and computer services suppliers. Furthermore, the technical and social division of labour during manufacturing production has largely inspired the rise of knowledge-intensive business services (KIBS) [44], which tend to be IT-intensive and based on social and institutional knowledge [43]. Meanwhile, products from the traditional professional services, computer, R&D and engineering services industries are mostly intangible services that require continuous interaction with both customers and suppliers [45]. As a result, firms in the manufacturing and services sectors have become closely related to IT services, which explains why industries in both the manufacturing and services sectors in York suffered severe economic losses when the IT services industry was made non-functional due to the flooding.

## 5. Conclusion

In this paper we have focused on the so-called 'Christmas' flood in York (UK), 2015. The case is special in the sense that little infrastructure was lost or damaged, while a single industry (IT services) was completely knocked out for a limited time. Due to these characteristics, the standard modelling techniques are no longer appropriate, one important reason being that, if used, many (model) parameters have to be newly calibrated or 'neglected', which severely affects the performance of the models in question.

HEM can be used to estimate the disaster-induced economic losses when one sector is completely knocked out and the other industries remained relatively unhurt. Because input–output tables at city level were not available, the Flegg and Webber 'scaling down' method was used as a first approximation. The outcomes were (§4) that the three-day shutdown of the IT services caused £3.24 million in economic losses. Of these, the IT services industry itself accounted for £1.83 million, and the remaining 44 other industries for £1.41 million out of total losses.

The Location Quotient has been widely applied in the regionalization of IO tables, and the Augmented Flegg Location Quotient is a variant of the Location Quotient family. This is adjusted to being based on the Flegg Location Quotient to take regional specialization into account, which makes it possible to scale national input coefficients upwards [46,47]. However, location capitals are largely based on the following assumptions: (i) Assuming identical productivity per employee in each region; (ii) Assuming consumption per employee of the same products, which basically refers to identical consumer preference; and (iii) Excluding cross hauling between regions as consumption would be fully satisfied by local supply if the region is the exporter for the given commodity [48,49]. Of these, most criticism is on the identical consumer preference which might seriously underestimate the interregional trade. The Cross-Hauling Adjusted Regionalization Method (CHARM) was designed to estimate the amount of cross-hauling, but it only is applicable to a type A IO table where imports are incorporated into the national transaction table [50]. But our study is based on the technology coefficients and final demand derived from a type B IO table (UK IO table), which would not be affected by trade estimates. Therefore, the cross-hauling problem in the LQ approach was omitted in this case.

We would like to point out that further research along this line is certainly required. First, the present study used an original HEM approach, which means that both backward and forward linkages of an industry were simply removed from the economy. This was motivated in terms of the industry in question (IT) being for a limited time and to a very large degree isolated from the rest of the economy (thereby bringing HEM use in line with recent criticism). Nonetheless, there can be further differentiation between an industry's internal and external linkages, as suggested by Cella [21]. Indeed, as criticized by Oosterhaven [34, p. 8], the original HEM approach considers

neither the substitutions and replacement possibilities through imports nor the higher order backward impacts of this disappearance and forward impacts of the secession of these sales upon the purchasing industries. While we agree that such criticism holds for most 'normal' flooding events, the York flood case reveals distinctive characteristics in both duration and nature of affected services, which validate the applicability of HEM. Second, possible changes in final demand were not considered in the present study due to the relatively short time period of the outage. Additionally, due to the lack of daily data on household expenditure in York, we could not specifically detect the exact value of sales during the three-day IT outage. In this respect, the current study opens up new research questions when applying the HEM once additional accurate data on daily household consumption or city-level IO tables become available.

All in all, we have been dealing with a very special case (one industry, three days, Christmas period). Nonetheless, this was a real case happening in the real world. In our view, this illustrates that it may be worthwhile to focus on selected characteristics of disasters and see if specialized input–output based models can come up with the most appropriate framework to analyse these cases. There may be cases where such specialized I-O models do perform very well.

Data accessibility. The data that support the findings of this study are available in https://figshare.com/s/f69104bae87b6bda2c80 and the electronic supplementary material.

Authors' contribution. D.G. and Y.X. designed the study and Y.X. carried them out. Y.X. conducted the calculation and prepared the manuscript with contributions from all co-authors.

Competing interests. The authors declare that they have no conflict of interest.

Funding. This study was supported by the National Key R&D Program of China (2016YFA0602604), the National Natural Science Foundation of China (41629501, 71533005), the Chinese Academy of Engineering (2017-ZD-15-07), the UK Natural Environment Research Council (NE/N00714X/1 and NE/P019900/1), the Economic and Social Research Council (ES/L016028/1), the Royal Academy of Engineering (UK-CIAPP/425) and the British Academy (NAFR2180103 and NAFR2180104).

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
