## [Reviewer comments · Proceedings. Mathematical, Physical, and Engineering Sciences]

Review History

RSPA-2018-0871.R0 (Original submission)

Review form: Referee 1

Is the manuscript an original and important contribution to its field?

Yes

Is the paper of sufficient general interest?

Yes

Is the overall quality of the paper suitable?

Yes

Quality of the paper

A good paper worth publishing in Proceedings.

Can the paper be shortened without overall detriment to the main message?

No

Do you think some of the material would be more appropriate as an electronic appendix?

No

For papers with colour figures – is colour essential?

No

If there is supplementary material, is this adequate and clear?

Not applicable

Are there details of how to obtain materials and data, including any restrictions that may apply?

Not applicable

Do you have any ethical concerns with this paper?

No

Recommendation?

Major revision is needed (please make suggestions in comments)

Comments to the Author(s)

General comments

The manuscript is well structured and written, I had no problem understanding everything. The figures and tables are informative and appropriate. I agree with the authors in that the York flood provides an interesting challenge, because the outage lasted only three days.

The HEM method has a long tradition, with major contributions from Dietzenbacher and van der Linden 1993; Miller and Lahr 2001; Dietzenbacher and Lahr 2013.

With regard to the regionalisation of the table, it is good that the authors try different methods. As far as I can see they use a number of location quotient variants. But what about cross-hauling? Kronenberg 2009 has developed the CHARM method which is not included. There could be major differences arising to the authors' results when CHARM variants are used, at least that is what comes out of analyses by Flegg and Tohmo 2012; Flegg et al. 2014; Lenzen et al. 2017. Then, why were no gravity approaches tested? Gallego and Lenzen 2009 have shown for Australia that gravity techniques improve location quotients.

Then, the authors scale final demand by 3/365 to arrive at 3-day effects. But in here they assume that final demand is temporally uniform across a whole year, which is unlikely to be the case, especially in the UK where heating, food choices etc are highly seasonal, and therefore food suppliers and utilities would experience seasonal variations of IT use. This issue is especially aggravated by the fact that the flood occurred during the Christmas season, where an unusually high use of IT could be expected caused by the Christmas shopping sprees.

A major shortcoming is though that the method does not account for downstream effects, and the authors acknowledge this in Section 2.2, where they discuss Oosterhaven's criticism. My question here would be: What has Oosterhaven himself recommended to include downstream effects? And what – if anything at all – could the authors take on board? After all, an IT outage will have major effects on any business that relies on these IT inputs. My gut feeling is that the backward effects may be rather uninteresting compared to the major forward effects. This is my main concern regarding the manuscript.

Minor comments

- Eq. 1 - in z_{ij} , z must be in italics. Similar eq. 10.
- Section 2.1: Please insert and discuss also Schulte in den Bäumen et al. 2015

References

- Dietzenbacher, E. and M.L. Lahr (2013) Expanding extractions. *Economic Systems Research* 25, 341-360.
- Dietzenbacher, E. and J.A. van der Linden (1993) The regional extraction method: EC input-output comparisons. *Economic Systems Research* 5, 185-206.
- Flegg, A.T., Y. Huang and T. Tohmo (2014) Cross-hauling and regional input-output tables: the case of the province of Hubei, China. *Economics Working Paper* 1310, <https://www2.uwe.ac.uk/faculties/BBS/BUS/Research/Economics13/1310.pdf>, Bristol, UK, University of the West of England.
- Flegg, A.T. and T. Tohmo (2012) A comment on Tobias Kronenberg's "Construction of regional input-output tables using non-survey methods: the role of cross-hauling". *International Regional Science Review* 35.
- Gallego, B. and M. Lenzen (2009) Estimating generalised regional input-output systems: A case study of Australia. In: M. Ruth and B. Davíðsdóttir (eds.) *The Dynamics of Regions and Networks in Industrial Ecosystems* Boston, MA, USA, Edward Elgar Publishing 55-82.
- Kronenberg, T. (2009) Construction of regional input-output tables using non-survey methods - the role of cross-hauling. *International Regional Science Review* 32, 40-64.
- Lenzen, M., A. Geschke, A. Malik, J. Fry, J. Lane, T. Wiedmann, S. Kenway, K. Hoang and A. Cadogan-Cowper (2017) New multi-regional input-output databases for Australia - enabling timely and flexible regional analysis. *Economic Systems Research* 29, in press.
- Miller, R.E. and M.L. Lahr (2001) A taxonomy of extractions. In: M.L. Lahr and R.E. Miller (eds.) *Regional Science Perspectives in Economic Analysis*. Amsterdam, Netherlands, Elsevier, 407-441.
- Schulte in den Bäumen, H., J. Többen and M. Lenzen (2015) Labour forced impacts and production losses due to the 2013 flood in Germany. *Journal of Hydrology* 527, 142-150.

Review form: Referee 2

Is the manuscript an original and important contribution to its field?

No

Is the paper of sufficient general interest?

Yes

Is the overall quality of the paper suitable?

No

Quality of the paper

A paper that may be acceptable after major revision.

Can the paper be shortened without overall detriment to the main message?

Yes

Do you think some of the material would be more appropriate as an electronic appendix?

No

For papers with colour figures – is colour essential?

No

If there is supplementary material, is this adequate and clear?

No

Are there details of how to obtain materials and data, including any restrictions that may apply?

Yes

Do you have any ethical concerns with this paper?

No

Recommendation?

Major revision is needed (please make suggestions in comments)

Comments to the Author(s)

The paper is about an interesting and timely topic. However, the paper should have more scientific rigour, especially in the language (e.g. Sec. 1.1). Major re-editing is expected during the review; other main issues include:

- validation: has the method being validated?

- limitation: in the abstract some limitation are mentioned, but this are not fully clarified in the main text

- implication: why this study is important and how can contribute to decision-making/robustness (the typical “so what?”)?

- future studies: could this study be repeated? Does the “specialty” of the case study be a problem for reproducing or transferring it?

Some specific notes are also detailed below.

P1 L 29-30: alternative...method?; Hypothetical Extraction Method

P2 L9: Boxing Day is on the 26 December?

P2 L44-46: I think authors should specify the kind of losses are meant; for example, during 25-26 December all shops are closed and I don't see much shopping lost. If other type of transactions are meant, this should be clarified.

P2 L53: Hypothetical Extraction Method

P3 L25-27: I would remove the two sentences

P3 L 41-42: I would remove the sentence

P4 L 4-11: I would keep the Ericson example much shorter, just focus on the loss

P4 L 12-14: authors should specify the “international manufacturer”, the loss and the type of incident

P4 L37-54: “indirect losses” denote the wider disruption to services, including all urban networks (power grid, drainage, transport). The literature review can be expanded to include e.g. (Thieken et al., 2008); Aerts et al. (2013); Pregnolato et al (2016)

P4 L 57-58: simplify the sentence e.g. “Post-disaster situations vary.”; I would specify what it is meant by “normal” flood

P8 L 11-19: I would integrate this sentences between brackets into the main text

P13 L 14: see above, transactions during the 25-26.12 are limited because of shops closure

P13 L 51-57: transform the “million decimals” (e.g. £0.075m) into a £75k; also at P14

P14 L21: clarify how you results are “in line” with other studies (e.g. compare numbers)

P14 L42: just “Conclusion”

P14 L 54-55: start the sentence with “Hypothetical Extraction Method”

P15 L3-4: has the results being validated somehow?

P15 L 7-24: could this be part of the discussion?

P15 L 26-33: I would remove the entire paragraph

Fig. 1: it would be useful to see also the £ lost in addition to %

Fig 2: axis titles are missing. Again, not used the decimals of millions if none of the losses arrive to one million or more

Aerts, J., Botzen, W., Bowman, M., Dircke, P. and Ward, P. (2013a) *Climate Adaptation and Flood Risk in Coastal Cities*. Taylor and Francis. Available at:

<http://ncl.ebib.com/patron/FullRecord.aspx?p=1576080>

Thieken, A.H., Ackermann, V., Elmer, F., Kreibich, H., Kuhlmann, B., Kunert, U., Maiwald, H., Merz, B., Muller, M., Piroth, K., Schwarz, J., Schwarze, R., Seifert, I. and Seifert, J. (2008)

Proceedings of the 4th International Symposium on Flood Defence, Toronto, Canada, 6–8 May 2008.

Pregolato, M., Ford, A., Robson, C., Glenis, V., Barr, S. and Dawson, R. (2016) 'Assessing urban strategies for reducing the impacts of extreme weather on infrastructure networks', *Royal Society Open Science*, 3(5), pp. 1-15.

Decision letter (RSPA-2018-0871.R0)

29-Jan-2019

Dear Dr Meng

The Editor of Proceedings A has now received comments from referees on the above paper and would like you to revise it in accordance with their suggestions which can be found below (not including confidential reports to the Editor).

Please submit a copy of your revised paper within four weeks - if we do not hear from you within this time then it will be assumed that the paper has been withdrawn. In exceptional circumstances, extensions may be possible if agreed with the Editorial Office in advance.

Please note that it is the editorial policy of Proceedings A to offer authors one round of revision in which to address changes requested by referees. If the revisions are not considered satisfactory by the Editor, then the paper will be rejected, and not considered further for publication by the journal. In the event that the author chooses not to address a referee's comments, and no scientific justification is included in their cover letter for this omission, it is at the discretion of the Editor whether to continue considering the manuscript.

- Acknowledgements
- Funding statement

To revise your manuscript, log into <http://mc.manuscriptcentral.com/prsa> and enter your Author Centre, where you will find your manuscript title listed under "Manuscripts with Decisions." Under "Actions," click on "Create a Revision." Your manuscript number has been appended to denote a revision.

You will be unable to make your revisions on the originally submitted version of the manuscript. Instead, revise your manuscript and upload a new version through your Author Centre.

When submitting your revised manuscript, you will be able to respond to the comments made by the referee(s) and upload a file "Response to Referees" in "Section 6 - File Upload". Please use this to document how you have responded to the comments, and the adjustments you have made. In order to expedite the processing of the revised manuscript, please be as specific as possible in your response to the referee(s).

IMPORTANT: Your original files are available to you when you upload your revised manuscript. Please delete any unnecessary previous files before uploading your revised version.

When revising your paper please ensure that it remains under 28 pages long. In addition, any pages over 20 will be subject to a charge (£150 + VAT (where applicable) per page). Your paper has been ESTIMATED to be 14 pages.

Once again, thank you for submitting your manuscript to Proc. R. Soc. A and I look forward to receiving your revision. If you have any questions at all, please do not hesitate to get in touch.

Yours sincerely

Alice Power
Publishing Editor
Proceedings A
proceedingsa@royalsociety.org

on behalf of
Professor Guangtao Fu
Board Member
Proceedings A

Board Member Comments to Author(s):

This paper has been reviewed by two reviewers and a number of major issues have been raised, ranging from case study data, model validation, knock-on effects to implications for decision making. The authors are suggested to provide a detailed response to all the comments from the two reviewers.

In addition, I would recommend the authors to avoid footnotes and use proper references instead. I also strongly recommend the case study data should be made easily accessible, for example through a repository with DOI rather than contacting the authors.

Reviewer(s)' Comments to Author:

Referee 1 Comments to the Author(s):

The manuscript is well structured and written, I had no problem understanding everything. The figures and tables are informative and appropriate. I agree with the authors in that the York flood provides an interesting challenge, because the outage lasted only three days.

The HEM method has a long tradition, with major contributions from Dietzenbacher and van der Linden 1993; Miller and Lahr 2001; Dietzenbacher and Lahr 2013.

With regard to the regionalisation of the table, it is good that the authors try different methods. As far as I can see they use a number of location quotient variants. But what about cross-hauling? Kronenberg 2009 has developed the CHARM method which is not included. There could be major differences arising to the authors' results when CHARM variants are used, at least that is what comes out of analyses by Flegg and Tohmo 2012; Flegg et al. 2014; Lenzen et al. 2017. Then, why were no gravity approaches tested? Gallego and Lenzen 2009 have shown for Australia that gravity techniques improve location quotients.

Then, the authors scale final demand by 3/365 to arrive at 3-day effects. But in here they assume that final demand is temporally uniform across a whole year, which is unlikely to be the case, especially in the UK where heating, food choices etc are highly seasonal, and therefore food suppliers and utilities would experience seasonal variations of IT use. This issue is especially aggravated by the fact that the flood occurred during the Christmas season, where an unusually high use of IT could be expected caused by the Christmas shopping sprees.

A major shortcoming is though that the method does not account for downstream effects, and the authors acknowledge this in Section 2.2, where they discuss Oosterhaven's criticism. My question here would be: What has Oosterhaven himself recommended to include downstream effects? And what – if anything at all – could the authors take on board? After all, an IT outage will have major effects on any business that relies on these IT inputs. My gut feeling is that the backward effects may be rather uninteresting compared to the major forward effects. This is my main concern regarding the manuscript.

Minor comments

- Eq. 1 – in z_{ij} , z must be in italics. Similar eq. 10.
- Section 2.1: Please insert and discuss also Schulte in den Bäumen et al. 2015

References

- Dietzenbacher, E. and M.L. Lahr (2013) Expanding extractions. *Economic Systems Research* 25, 341-360.
- Dietzenbacher, E. and J.A. van der Linden (1993) The regional extraction method: EC input-output comparisons. *Economic Systems Research* 5, 185-206.
- Flegg, A.T., Y. Huang and T. Tohmo (2014) Cross-hauling and regional input-output tables: the case of the province of Hubei, China. *Economics Working Paper* 1310, <https://www2.uwe.ac.uk/faculties/BBS/BUS/Research/Economics13/1310.pdf>, Bristol, UK, University of the West of England.
- Flegg, A.T. and T. Tohmo (2012) A comment on Tobias Kronenberg's "Construction of regional input-output tables using non-survey methods: the role of cross-hauling". *International Regional Science Review* 35.
- Gallego, B. and M. Lenzen (2009) Estimating generalised regional input-output systems: A case study of Australia. In: M. Ruth and B. Davíðsdóttir (eds.) *The Dynamics of Regions and Networks in Industrial Ecosystems* Boston, MA, USA, Edward Elgar Publishing 55-82.
- Kronenberg, T. (2009) Construction of regional input-output tables using non-survey methods - the role of cross-hauling. *International Regional Science Review* 32, 40-64.

- Lenzen, M., A. Geschke, A. Malik, J. Fry, J. Lane, T. Wiedmann, S. Kenway, K. Hoang and A. Cadogan-Cowper (2017) New multi-regional input-output databases for Australia – enabling timely and flexible regional analysis. *Economic Systems Research* 29, in press.
- Miller, R.E. and M.L. Lahr (2001) A taxonomy of extractions. In: M.L. Lahr and R.E. Miller (eds.) *Regional Science Perspectives in Economic Analysis*. Amsterdam, Netherlands, Elsevier, 407-441.
- Schulte in den Bäumen, H., J. Többen and M. Lenzen (2015) Labour forced impacts and production losses due to the 2013 flood in Germany. *Journal of Hydrology* 527, 142-150.

Referee 2 Comments to the Author(s):

The paper is about an interesting and timely topic. However, the paper should have more scientific rigour, especially in the language (e.g. Sec. 1.1). Major re-editing is expected during the review; other main issues include:

- validation: has the method being validated?
- limitation: in the abstract some limitation are mentioned, but this are not fully clarified in the main text
- implication: why this study is important and how can contribute to decision-making/robustness (the typical “so what?”)?
- future studies: could this study be repeated? Does the “specialty” of the case study be a problem for reproducing or transferring it?

Some specific notes are also detailed below.

P1 L 29-30: alternative...method?; Hypothetical Extraction Method

P2 L9: Boxing Day is on the 26 December?

P2 L44-46: I think authors should specify the kind of losses are meant; for example, during 25-26 December all shops are closed and I don't see much shopping lost. If other type of transactions are meant, this should be clarified.

P2 L53: Hypothetical Extraction Method

P3 L25-27: I would remove the two sentences

P3 L 41-42: I would remove the sentence

P4 L 4-11: I would keep the Ericsson example much shorter, just focus on the loss

P4 L 12-14: authors should specify the “international manufacturer”, the loss and the type of incident

P4 L37-54: “indirect losses” denote the wider disruption to services, including all urban networks (power grid, drainage, transport). The literature review can be expanded to include e.g. (Thieken et al., 2008); Aerts et al. (2013); Pregolato et al (2016)

P4 L 57-58: simplify the sentence e.g. “Post-disaster situations vary.”; I would specify what it is meant by “normal” flood

P8 L 11-19: I would integrate this sentences between brackets into the main text

P13 L 14: see above, transactions during the 25-26.12 are limited because of shops closure

P13 L 51-57: transform the “million decimals” (e.g. £0.075m) into a £75k; also at P14

P14 L21: clarify how you results are “in line” with other studies (e.g. compare numbers)

P14 L42: just “Conclusion”

P14 L 54-55: start the sentence with “Hypothetical Extraction Method”

P15 L3-4: has the results being validated somehow?

P15 L 7-24: could this be part of the discussion?

P15 L 26-33: I would remove the entire paragraph

Fig. 1: it would be useful to see also the £ lost in addition to %

Fig 2: axis titles are missing. Again, not used the decimals of millions if none of the losses arrive to one million or more

Aerts, J., Botzen, W., Bowman, M., Dircke, P. and Ward, P. (2013a) Climate Adaptation and Flood Risk in Coastal Cities. Taylor and Francis. Available at:

<http://ncl.ebib.com/patron/FullRecord.aspx?p=1576080>

Thieken, A.H., Ackermann, V., Elmer, F., Kreibich, H., Kuhlmann, B., Kunert, U., Maiwald, H., Merz, B., Muller, M., Piroth, K., Schwarz, J., Schwarze, R., Seifert, I. and Seifert, J. (2008)

Proceedings of the 4th International Symposium on Flood Defence, Toronto, Canada, 6–8 May 2008.

Pregolato, M., Ford, A., Robson, C., Glenis, V., Barr, S. and Dawson, R. (2016) 'Assessing urban strategies for reducing the impacts of extreme weather on infrastructure networks', Royal Society Open Science, 3(5), pp. 1-15.

Author's Response to Decision Letter for (RSPA-2018-0871.R0)

See Appendix A.

RSPA-2018-0871.R1 (Revision)

Review form: Referee 1

Is the manuscript an original and important contribution to its field?

Yes

Is the paper of sufficient general interest?

Yes

Is the overall quality of the paper suitable?

Yes

Quality of the paper

A good paper worth publishing in Proceedings.

Can the paper be shortened without overall detriment to the main message?

No

Do you think some of the material would be more appropriate as an electronic appendix?

No

For papers with colour figures – is colour essential?

No

If there is supplementary material, is this adequate and clear?

Not applicable

Are there details of how to obtain materials and data, including any restrictions that may apply?

Not applicable

Do you have any ethical concerns with this paper?

No

Recommendation?

Accept as is

Comments to the Author(s)

The authors have responded well to my comments. The paper can be published.

Review form: Referee 2

Is the manuscript an original and important contribution to its field?

Yes

Is the paper of sufficient general interest?

Yes

Is the overall quality of the paper suitable?

Yes

Quality of the paper

A good paper worth publishing in Proceedings.

Can the paper be shortened without overall detriment to the main message?

No

Do you think some of the material would be more appropriate as an electronic appendix?

No

For papers with colour figures – is colour essential?

No

If there is supplementary material, is this adequate and clear?

Yes

Are there details of how to obtain materials and data, including any restrictions that may apply?

Not applicable

Do you have any ethical concerns with this paper?

No

Recommendation?

Accept as is

Comments to the Author(s)

The article is ready for publication

Decision letter (RSPA-2018-0871.R1)

Dear Dr Meng

On behalf of the Editor, I am pleased to inform you that your manuscript entitled "Assessing the Economic Impacts of IT Service Shutdown during the York Flood of 2015 in the UK" has been accepted in its final form for publication in Proceedings A.

Our Production Office will be in contact with you in due course. You can expect to receive a proof of your article soon. Please contact the office to let us know if you are likely to be away from e-mail in the near future. If you do not notify us and comments are not received within 5 days of sending the proof, we may publish the paper as it stands.

Open access

You are invited to opt for open access, our author pays publishing model. Payment of open access fees will enable your article to be made freely available via the Royal Society website as soon as it is ready for publication. For more information about open access please visit http://royalsocietypublishing.org/site/authors/open_access.xhtml. The open access fee for this journal is £1700/\$2380/€2040 per article. VAT will be charged where applicable.

Note that if you have opted for open access then payment will be required before the article is published – payment instructions will follow shortly. If you wish to opt for open access then please inform the editorial office (proceedingsa@royalsociety.org) as soon as possible.

Your article has been estimated as being 14 pages long. Our Production Office will inform you of the exact length at the proof stage.

Proceedings A levies charges for articles which exceed 20 printed pages. (based upon approximately 540 words or 2 figures per page). Articles exceeding this limit will incur page charges of £150 per page or part page, plus VAT (where applicable).

Under the terms of our licence to publish you may post the author generated postprint (ie. your accepted version not the final typeset version) of your manuscript at any time and this can be made freely available. Postprints can be deposited on a personal or institutional website, or a recognised server/repository. Please note however, that the reporting of postprints is subject to a media embargo, and that the status the manuscript should be made clear. Upon publication of the definitive version on the publisher's site, full details and a link should be added.

You can cite the article in advance of publication using its DOI. The DOI will take the form: 10.1098/rspa.XXXX.YYYY, where XXXX and YYYY are the last 8 digits of your manuscript number (eg. if your manuscript number is RSPA-2017-1234 the DOI would be 10.1098/rspa.2017.1234).

For tips on promoting your accepted paper see our blog post: <https://blogs.royalsociety.org/publishing/promoting-your-latest-paper-and-tracking-your-results/>

Thank you for your submission. On behalf of the Editors of the journal, we look forward to your continued contributions to the Journal.

Best wishes
Alice Power
Proceedings A Editorial Office

proceedingsa@royalsociety.org

on behalf of
Professor Guangtao Fu
Board Member
Proceedings A

Reviewer(s)' Comments to Author:

Referee: 1

Comments to the Author(s)
The authors have responded well to my comments. The paper can be published.

Referee: 2

Comments to the Author(s)
The article is ready for publication

Appendix A

Board Member Comments to Author(s):

This paper has been reviewed by two reviewers and a number of major issues have been raised, ranging from case study data, model validation, knock-on effects to implications for decision-making. It has been suggested that the authors provide a detailed response to all the comments from the two reviewers.

In addition, I would recommend the authors to avoid footnotes and use proper references instead. I also strongly recommend the case study data should be made easily accessible, for example through a repository with DOI rather than contacting the authors.

Thank you very much. We have removed the footnotes and replaced them with proper references.

Reviewer(s)' Comments to Author:

Referee 1 Comments to the Author(s):

The manuscript is well structured and written, I had no problem understanding everything. The figures and tables are informative and appropriate. I agree with the authors in that the York flood provides an interesting challenge, because the outage lasted only three days.

Thank you for acknowledging the potential interest of our selected case. We strongly agree that the York flood offers an interesting and special case that invalidates existing flood modelling techniques because of little damage caused to the physical infrastructure.

The HEM method has a long tradition, with major contributions from Dietzenbacher and van der Linden 1993; Miller and Lahr 2001; Dietzenbacher and Lahr 2013.

Thank you for your comments. We agree with you and appreciate the significant contributions made by existing literature to the HEM method. We have incorporated your mentioned references in our revised manuscript (please see line 185, 159 and 161).

With regard to the regionalisation of the table, it is good that the authors try different methods. As far as I can see they use a number of location quotient variants. But what about cross-hauling? Kronenberg 2009 has developed the CHARM method which is not included. There could be major differences arising to the authors' results when CHARM variants are used, at least that is what comes out of analyses by Flegg and Tohmo 2012; Flegg et al. 2014; Lenzen et al. 2017. Then, why were no gravity approaches tested? Gallego and Lenzen 2009 have shown for Australia that gravity techniques improve location quotients.

We appreciate the reviewer's comments considering the possible effects from cross-hauling. We do agree cross-hauling would have significant impacts in trade flow estimates. However, in our study, trade flows were not used, because our HEM method only focuses on technological coefficients (A) and final demand (F). Given that the UK national IO table is the type B IO table we used here, domestic and import matrix are separated, and therefore, our LQ approach to estimate A and then F for York is only based on the domestic matrix, excluding trade. Therefore, whether or not trade is precisely estimated does not matter in terms of our purpose in this case. If our purpose here would have been to construct a solid city-level IO table, it is definitely necessary to consider CHARM or use a gravity theory-based model. However, it requires other city-level trade data as sample data which are not available. To clarify the method, we added a brief description in section 5 to discuss methods you mentioned and stated why it is not applied in this study, as shown below:

"The Location Quotient has been widely applied in the regionalisation of IO tables, and the Augmented Flegg Location Quotient is a variant of the Location Quotient family. This is adjusted to be based on the Flegg Location Quotient to take regional specialisation into account, which make it possible to scale national input coefficients upwards (Flegg et al. 2016; Jahn 2017). However, location capitals is

largely based on the following assumptions: 1. Assuming identical productivity per employee in each region; 2. Assuming consumption per employee of the same products, which basically refers to identical consumer preference; and 3. Excluding cross hauling between regions as consumption would be fully satisfied by local supply if the region is the exporter for the given commodity (Riddington et al. 2006; Zheng et al. 2019). Of these, most criticism is on the identical consumer preference which might seriously underestimate the interregional trade. The Cross-Hauling Adjusted Regionalization Method (CHARM) was designed to estimate the amount of cross-hauling, but it only is applicable to a type A IO table where imports are incorporated into the national transaction table (Flegg et al. 2015). But our study is based on the technology coefficients and final demand derived from a type B IO table (UK IO table), which would not be affected by trade estimates. Therefore, the cross-hauling problem in the LQ approach was omitted in this case.”

Then, the authors scale final demand by 3/365 to arrive at 3-day effects. But in here they assume that final demand is temporally uniform across a whole year, which is unlikely to be the case, especially in the UK where heating, food choices etc are highly seasonal, and therefore food suppliers and utilities would experience seasonal variations of IT use. This issue is especially aggravated by the fact that the flood occurred during the Christmas season, where an unusually high use of IT could be expected caused by the Christmas shopping sprees.

We thank the reviewer for these constructive comments. We agree with you that our assumption of averaged final demand cannot fully represent the seasonal variations in consumption. Therefore, in the revised manuscript, we have added an upper boundary of the results following the same methods but employed different numbers for final demand during the three-day IT outages. Due to the lack of daily sales data for York, we assumed the same monthly trend in household expenditure as the UK. According to data from the Office for National Statistics (2016), household expenditure on food, drink and tobacco, clothing and footwear and other household goods during December are 16%, 42% and 31% higher than those of other non-Christmas months during 2015. Therefore, we adjusted the original three days' final demand that was calculated from the IO table. The upper boundary of economic loss considering the Christmas consumption peak is estimated as £4.23 million (please see lines 393-402. While we believe such estimates still contain uncertainties, we believe the provision of an estimate range in economic loss is the best we can do due to the lack of detailed seasonal transaction data at hand.

A major shortcoming is though that the method does not account for downstream effects, and the authors acknowledge this in Section 2.2, where they discuss Oosterhaven's criticism. My question here would be: What has Oosterhaven himself recommended to include downstream effects? And what – if anything at all – could the authors take on board? After all, an IT outage will have major effects on any business that relies on these IT inputs. My gut feeling is that the backward effects may be rather uninteresting compared to the major forward effects. This is my main concern regarding the manuscript.

We thank the reviewer for these insightful comments. For disasters lasting longer, we agree with the criticism raised by Oosterhaven. What he basically argues is that the extraction behaviour in the HEM method is based on the assumption that all downstream demand for a sector's intermediate sales simply and completely disappear after the incident. In contrast, he thinks such downstream demand for intermediate sales will not disappear, but rather, seeks for substitutions elsewhere. Also, the HEM neither measures the higher order backward impacts of this disappearance, nor the forward impacts of the secession of these sales upon the purchasing industries. This is, of course, the case for any disaster that lasts longer and when sectors have enough time to adapt. However, we suggest his criticism does not hold for our special York flood case. This is because the IT outage resulting from the flood only lasts for three days, which provide insufficient time for economic sectors to adapt or seek for other substitutions. This is also because of the special nature of IT services produced by local carriers that invalidate the replacement or substitutions of imports.

Minor comments

- Eq. 1 – in z_{ij} , z must be in italics. Similar eq. 10.

Thank you very much. We have changed them into italics accordingly.

- Section 2.1: Please insert and discuss also Schulte in den Bäumen et al. 2015

Thank you very much. We have inserted and discussed the recommended reference on line 111-116.

References

- Dietzenbacher, E. and M.L. Lahr (2013) Expanding extractions. *Economic Systems Research* 25, 341-360.
- Dietzenbacher, E. and J.A. van der Linden (1993) The regional extraction method: EC input-output comparisons. *Economic Systems Research* 5, 185-206.
- Flegg, A.T., Y. Huang and T. Tohmo (2014) Cross-hauling and regional input-output tables: the case of the province of Hubei, China. *Economics Working Paper 1310*, <https://www2.uwe.ac.uk/faculties/BBS/BUS/Research/Economics13/1310.pdf>, Bristol, UK, University of the West of England.
- Flegg, A.T. and T. Tohmo (2012) A comment on Tobias Kronenberg's "Construction of regional input-output tables using non-survey methods: the role of cross-hauling". *International Regional Science Review* 35.
- Gallego, B. and M. Lenzen (2009) Estimating generalised regional input-output systems: A case study of Australia. In: M. Ruth and B. Davíðsdóttir (eds.) *The Dynamics of Regions and Networks in Industrial Ecosystems* Boston, MA, USA, Edward Elgar Publishing 55-82.
- Kronenberg, T. (2009) Construction of regional input-output tables using non-survey methods - the role of cross-hauling. *International Regional Science Review* 32, 40-64.
- Lenzen, M., A. Geschke, A. Malik, J. Fry, J. Lane, T. Wiedmann, S. Kenway, K. Hoang and A. Cadogan-Cowper (2017) New multi-regional input-output databases for Australia – enabling timely and flexible regional analysis. *Economic Systems Research* 29, 275-295.
- Miller, R.E. and M.L. Lahr (2001) A taxonomy of extractions. In: M.L. Lahr and R.E. Miller (eds.) *Regional Science Perspectives in Economic Analysis*. Amsterdam, Netherlands, Elsevier, 407-441.
- Schulte in den Bäumen, H., J. Többen and M. Lenzen (2015) Labour forced impacts and production losses due to the 2013 flood in Germany. *Journal of Hydrology* 527, 142-150.

Referee 2 Comments to the Author(s):

The paper is about an interesting and timely topic. However, the paper should have more scientific rigour, especially in the language (e.g. Sec. 1.1). Major re-editing is expected during the review; other main issues include:

- validation: has the method being validated?

Thank you for this question. The Hypothetical Extraction Method has a long tradition in disaster risk analysis. It was first proposed to estimate the relative importance of certain industries for an entire economy by Paelinck et al. (1965) and Strassert (1968), and later re-formulated by Meller and Marfan (1981) and Cella (1984). The method was used to estimate the effects of the extraction on other industries and on the wider economic system when an industry is hypothetically eliminated from the economic system, and the difference between the output level of the other industries before and after the extraction reflects the linkages between the extracted industry and the rest of the economy. While the method was once criticized by Oosterhaven (2017) in ignoring the possibility of substitutions and downstream demand for intermediate sales, we think the York flood is a special case that validates the method because the flood lasted only for three days and the services could not be replaced by sources elsewhere.

Paelinck, J., de Caemel, J. and Degueldre, J. (1965) Analyse Quantitative de Certaines Phénomènes du Développement Régional Polarisé: Essai de Simulation Statique d'Itinéraires de Propagation'. In: Bibliothèque de l'Institut de Science économique. No. 7. Problèmes de Conversion économique: Analyses Théoriques études Appliquées, Génin, Paris, 341-387.

Strassert, G. (1968) Zur bestimmung strategischer Sektoren mit Hilfe von input-output Modellen. Jahrbücher für Nationalökonomie und Statistik, 182, 211-215.

Meller, P. and Marfan, M. (1981) Small and large industry: employment generation, linkages, and key sectors. Economic Development and Cultural Change, 29 (2), 263-274.

Cella, G., 1984. The input-output measurement of interindustry linkages. Oxford Bulletin of Economics and Statistics, 46, 73-84.

Oosterhaven, J. (2017) On the limited usability of the inoperability IO model. *Economic Systems Research*, 29(3), 452-461.

- limitation: in the abstract some limitations are mentioned, but these are not fully clarified in the main text

Thank you for the constructive comments. We have specified our discussion on limitation in section 5 (Conclusion) as

'We would like to point out that further research along this line is certainly required. First, the present study utilized an original HEM approach, which means that both backward and forward linkages of an industry were simply removed from the economy. This was motivated in terms of the industry in question (IT) being for a limited time and to a very large degree isolated from the rest of the economy (thereby bringing HEM use in line with recent criticism). Nonetheless, there can be further differentiation between an industry's internal and external linkages, as suggested by Cella (1984). Indeed, as criticized by Oosterhaven (2017, p.8), the original HEM approach considers neither the substitutions and replacement possibilities through imports, nor the higher order backward impacts of this disappearance and forward impacts of the secession of these sales upon the purchasing industries. While we agree that such criticism holds for most 'normal' flooding events, the York flood case reveals distinctive characteristics in both duration and nature of affected services, which validate the applicability of HEM. Second, possible changes in final demand were not considered in the present study due to the relatively short time period of the outage. Additionally, due to the lack of daily data on household expenditure in York, we could not specifically detect the exact value of sales during the three-day IT outages. In this respect, the current study opens up new research questions when applying the HEM once additional accurate data on daily household consumption or city-level IO tables become available.'

- implication: why this study is important and how can it contribute to decision-making/robustness (the typical "so what?")?

Thank you for your question. As we discussed in the Introduction, traditional ways of flood and disaster modelling become less useful for special cases, such as the York flood case here, for several reasons. Most important is that existing flood and disaster modelling efforts rely heavily on quantifying the damages to infrastructures as a direct and tangible consequence of flooding. However, not all flooding events will cause damage to infrastructure and this makes it difficult to implement standard ways of disaster modelling. It appears challenging to estimate the flood

induced indirect and intangible costs from soft services shutdowns, such as the IT service here. Therefore, our contribution to decision-making/robustness here is to provide an alternative and feasible tool for these 'special' cases, which have similar features in short duration and impact on soft services that both invalidate traditional disaster modelling tools.

- future studies: could this study be repeated? Does the "specialty" of the case study be a problem for reproducing or transferring it?

Thank you for your questions. While we acknowledge that the York flood case is a special case compared with other major flooding events, it is not a unique case in the context of a growing number of disastrous events. The York flood case is special in the sense that it lasted only for a short period of time – three days and affected only soft services – IT services, which cannot be replaced immediately. Traditional risk assessment tools are invalidated by such a case because they either measure the direct economic loss resulting from the damages to physical infrastructures, or measure the indirect economic loss cascading along the downstream and upstream production chains. Therefore, our aim here is to provide an alternative tool – the Hypothetical Extraction Method - for these special disaster events with similar features when other risk modelling tools can no longer be used.

Some specific notes are also detailed below.

P1 L 29-30: alternative...method?; Hypothetical Extraction Method

Thank you. We have reedited the sentence.

P2 L9: Boxing Day is on the 26 December?

Thank you very much. We have corrected the mistake.

P2 L44-46: I think authors should specify the kind of losses that are meant; for example, during 25-26 December all shops are closed and I don't see much shopping lost. If other type of transactions are meant, this should be clarified.

Thank you very much. We have clarified this point accordingly.

P2 L53: Hypothetical Extraction Method

Thank you. We have reedited according to your comments.

P3 L25-27: I would remove the two sentences

Thank you. We have removed the sentences accordingly.

P3 L 41-42: I would remove the sentence

Thank you. We have removed the sentence accordingly.

P4 L 4-11: I would keep the Ericsson example much shorter, just focus on the loss

Thank you very much. We have simplified the example according to your comments.

P4 L 12-14: authors should specify the “international manufacturer”, the loss and the type of incident

Thank you very much. We have clarified and specified the loss and the incident type according to your suggestions.

P4 L37-54: “indirect losses” denote the wider disruption to services, including all urban networks (power grid, drainage, transport). The literature review can be expanded to include e.g. (Thieken et al., 2008); Aerts et al. (2013); Pregolato et al (2016)

Thank you very much. We have inserted and discussed the recommended reference in the Literature Review section.

P4 L 57-58: simplify the sentence e.g. “Post-disaster situations vary.”; I would specify what is meant by “normal” flood

Thank you. We have simplified the sentence to clarify what we meant by ‘normal’ floods.

P8 L 11-19: I would integrate these sentences between brackets into the main text

Thank you. We have made the change accordingly.

P13 L 14: see above, transactions during the 25-26.12 are limited because of shops closure

Thank you. We have made the change according to your comments.

P13 L 51-57: transform the “million decimals” (e.g. £0.075m) into a £75k; also at P14

Thank you very much. We have changed the unit according to your comments.

P14 L21: clarify how your results are “in line” with other studies (e.g. compare numbers)

We agree that 'in line' is not an appropriate word here. We have clarified and changed it into '*These results reveal the heavy reliance on the supply of IT services in their operations and production. This, on the other hand, also acknowledges and confirms the recent remarkable growth in IT outsourcing in the UK.*'

P14 L42: just "Conclusion"

Thank you. We have made the change accordingly.

P14 L 54-55: start the sentence with "Hypothetical Extraction Method"

Thank you. We have made the change accordingly.

P15 L3-4: has the results being validated somehow?

Thank you for your comments. To validate our results and consider the extensive transaction volumes that may occur during the Christmas period, we have added an upper boundary to the results following the same methods but have employed different numbers for final demand during the three-day IT outages. Due to the lack of daily sales data for York, we assumed the same monthly trend in household expenditure as in the UK. According to data from the Office for National Statistics (2016), household expenditure on food, drink and tobacco, clothing and footwear and other household goods during December are 16%, 42% and 31% higher than those of other non-Christmas months during 2015. Therefore, we adjusted the original three days' final demand that was calculated from the IO table. The upper boundary of economic loss considering the Christmas consumption peak is estimated as £4.23 million.

P15 L 7-24: could this be part of the discussion?

Thank you very much for this comment. We have expanded this part of the discussion to provide an upper bound, as well as the discussion on p7 under *section 2.2 Applications of the hypothetical extraction method (HEM)*.

P15 L 26-33: I would remove the entire paragraph Fig. 1: it would be useful to see also the £ lost in addition to % Fig 2: axis titles are missing. Again, do not use the decimals of millions if none of the losses arrive to one million or more

Thank you for this comment. In order to provide both the broad and sector-specific picture regarding the economic losses, we decided to keep the Figure1. However, we integrated the paragraphs in section 4 Results to make them more consistent.

Aerts, J., Botzen, W., Bowman, M., Dircke, P. and Ward, P. (2013a) *Climate Adaptation and Flood Risk in Coastal Cities*. Taylor and Francis. Available at: <http://ncl.ebib.com/patron/FullRecord.aspx?p=1576080>

Thieken, A.H., Ackermann, V., Elmer, F., Kreibich, H., Kuhlmann, B., Kunert, U., Maiwald, H., Merz, B., Muller, M., Piroth, K., Schwarz, J., Schwarze, R., Seifert, I. and Seifert, J. (2008) *Proceedings of the 4th International Symposium on Flood Defence*, Toronto, Canada, 6–8 May 2008.

Pregolato, M., Ford, A., Robson, C., Glenis, V., Barr, S. and Dawson, R. (2016) 'Assessing urban strategies for reducing the impacts of extreme weather on infrastructure networks', *Royal Society Open Science*, 3(5), pp. 1-15.